# *H*_∞_ State-Feedback Control of Multi-Agent Systems with Data Packet Dropout in the Communication Channels: A Markovian Approach

**DOI:** 10.3390/e24121734

**Published:** 2022-11-28

**Authors:** Adrian-Mihail Stoica, Serena Cristiana Stoicu

**Affiliations:** Faculty of Aerospace Engineering, University Politehnica of Bucharest, 011061 Bucharest, Romania

**Keywords:** multi-agent systems, *H_∞_* type control, data packet dropout, Markovian models, coupled algebraic Riccati equations, iterative numerical methods

## Abstract

The paper presents an $H_{\infty }$ type control procedure for multi-agent systems taking into account possible data dropout in the communication network. The data dropout is modelled using a standard homogeneous Markov chain leading to an $H_{\infty }$ type control problem for stochastic multi-agent systems with Markovian jumps. The considered $H_{\infty }$ type criterion includes, besides the components corresponding to the attenuation condition of exogenous disturbance inputs, quadratic terms aiming to acquire the consensus between the agents. It is shown that in the case of identical agents, a state-feedback controller with Markov parameters may be determined solving two specific systems of Riccati equations whose dimension does not depend on the number of agents. Iterative procedures to solve such systems are also presented together with an illustrative numerical example.

## 1. Introduction

Multi-agent systems received a considerable interest in control engineering over the last decades due to their wide area of applications including terrestrial, maritime, aerial and space surveillance and monitoring missions. Some early developments and comprehensive surveys in this field may be found, for instance, in [1,2,3,4]. The design requirements for the control of multi-agent systems may be formulated from different perspectives, a lot of literature treating these topics being available these days. Although the review of the control design methodologies for multi-agent systems is beyond the purpose of this paper, one will mention however some monographs as [5,6,7,8], presenting different multi-agent control problems. An important aspect related the multi-agent systems is their distributed control which is characterised, in contrast with the centralised case, by the absence of a control decision-maker. Such formulations, primarily considered for agents with single and double integrators models (see, e.g., [9]) have been then investigated for more general linear systems, as in [10]. Consensus of nonlinear agents using output feedback has been analysed, for instance, in [11] and an algorithm based on feedback linearisation of nonlinear agents with output measurements may be found in [12]. In [13], a stability analysis from the perspective of a hybrid modelling is proposed using invariant sets and the Lyapunov stability theory. Consensus problems for Bernoulli networks have been considered for instance in [14] and for high-order multi-agent systems under deterministic and Markovian switching network topologies, in [15]. Other consensus problems in stochastic sense have been investigated, for instance, in [16] and in [17] under Markov switching networks for agents with first and second order dynamics. Many formulations of the control problems for multi-agent systems include optimisation criteria. As shown for instance, in [18], the complexity associated with the computation of distributed optimal controller significantly increases. In [19], the authors study the structural properties of optimal control problems with infinite-horizon linear-quadratic criteria by analysing the spatial structure of the solution to corresponding Lyapunov operator and Riccati equations. A problem of synthesising a distributed dynamic output feedback achieving $H_{\infty }$ performance is presented in [20]. In [21], a decentralised Markovian $H_{\infty }$ control problem is considered for first-order dynamics of the agents in presence of time-delay conditions; the optimal control is expressed in terms of the solutions of a system of linear matrix inequalities depending on the dynamics of the multi-agent system. An $H_{\infty }$ state-feedback consensus controller for undirected multi-agent systems is derived in [22] for a more general class of *N* identical agents; the complexity of the $H_{\infty }$ consensus problem is reduced by representing the problem by *N* number linear systems. In fact, optimal control problems of multi-agent systems remain a domain of interest due to the design and implementation complexity of the control laws, even in the case of identical agents (see, for instance, [22,23,24,25]). In [26], a linear-quadratic control problem for identical agents is considered, for which it is proved that the optimal solution depends on the number of agents and on the stabilising solutions of two Riccati equations having the same order as the agents dynamics. This conclusion shows that the computational requirements may be significantly reduced, especially for large-scale multi-agent systems. The research has been continued in [27] where the robustness properties of the decentralised linear-quadratic optimal controller are analysed.

More recently, an $H_{\infty }$ optimisation problem for identical stochastic linear models corrupted with multiplicative noises was formulated and solved in [28], aiming to provide disturbance attenuation performance together with robust stability with respect to parametric uncertainties in the agents models. Based on the state-feedback gains of the centralised state-feedback controller, a distributed controller depending on the adjacency matrix associated with the undirected graph of the communication network was obtained using the spectra of Lyapunov operators.

In this paper, the loss of links between agents is modelled by linear stochastic systems with Markovian jumps. The proposed methodology allows to consider different configurations of the network, each of them corresponding to a state of the Markov chain. Markov switching network models may be found in many papers between which one mentions [29,30,31]. The formulation and the developments presented in this paper considers stochastic models both for the agents and for the network. The $H_{\infty }$ cost function for the optimal control design includes, besides the expression for the attenuation of exogenous disturbance inputs, quadratic terms aiming to acquire the state consensus between the agents. The optimal $H_{\infty }$ state-feedback gains are expressed in terms of the stabilising solutions of two systems of coupled game-theoretic Riccati equations having the same order as the dynamics of a single agent. This coupling between the Riccati equations is typical in the optimal control of stochastic Markovian systems, depending on the elements of the stationary transition rate matrix.

The paper is organised as follows: the $H_{\infty }$ control problem for stochastic system with Markovian jumps is formulated and solved in Section 2. The optimal gains of the control law are expressed in terms of the stabilising solution of a system of coupled algebraic Riccati equations with indefinite sign. A convergent iterative algorithm to determine the stabilising solution of this system of coupled Riccati equations is also presented. The Section 3 analyses the case of multi-agent $H_{\infty }$ control. The main result of this section allows to determine the optimal control law for multi-agent systems solving two specific systems of coupled game-theoretic Riccati equations corresponding to the dimension of a single agent. In Section 4, the case of dropout data packages in the communication networks is discussed and illustrated by a numerical example for a large-scale multi-agent system with two states of the Markov chain. The paper ends with some concluding remarks.

## 2. $H_{\infty }$ Type Control for Stochastic Systems with Markovian Jumps; The Case of a Single Agent

Consider the linear stochastic system
(1)$$\begin{array}{c} \begin{array}{ccc} \dot{x}\left(t\right) & = & A\left(\eta \left(t\right)\right)x\left(t\right)+B_{1}\left(\eta \left(t\right)\right)w\left(t\right)+B_{2}\left(\eta \left(t\right)\right)u\left(t\right)\\ y_{1}\left(t\right) & = & C(\eta (t))x(t)+D(\eta (t))u(t)\\ y_{2}\left(t\right) & = & x(t) \end{array} \end{array}$$
where $x\in R^{n}$ denotes the state vector, $w\in R^{m_{1}}$ is an exogenous input, $u\in R^{m_{2}}$ stands for the control input, $y_{1}\in R^{p_{1}}$ is the quality output and $y_{2}\in R^{n}$ denotes the measured output. Throughout the paper η(t),t≥0 denotes a continuous Markov chain with the state space D={1,…,d} and with the probability transition matrix $P\left(t\right)=\left[p_{ij}\left(t\right)\right]=e^{\Pi t},\;i,j\in D,\;t\geq 0$ in which the stationary transition rate matrix of η is $\Pi =\left[\pi_{ij}\right]$ with $\sum_{j=1}^{d}\pi_{ij}=0,i\in D$ and $\pi_{ij}\geq 0$ if i≠j.

The triple {Ω,F,P} denotes a given probability space, E[x] stands for the expectation of the random variable *x*, $E\left[x|H\right]$ represents the conditional expectation of *x* with respect to the σ-algebra H⊂F and E[x|η(t)=i] is the conditional expectation with respect to the event η(t)=i. In the following developments it will be assumed that $C^{⊤}\left(i\right)D\left(i\right)=0$ and $D^{⊤}\left(i\right)D\left(i\right)=I_{m_{2}},\;\forall i\in D$. For invertible $D{\left(i\right)}^{⊤}D\left(i\right)$, i=1,…,d, if these assumptions are not accomplished one may perform the following change of the control variable *u*
$$\begin{array}{c} u\left(\eta \left(t\right)\right)=-{\left(D{\left(\eta \left(t\right)\right)}^{⊤}D\left(\eta \left(t\right)\right)\right)}^{-1}D{\left(\eta \left(t\right)\right)}^{⊤}C\left(\eta \left(t\right)\right)x\left(t\right)+{\left(D{\left(\eta \left(t\right)\right)}^{⊤}D\left(\eta \left(t\right)\right)\right)}^{-\frac{1}{2}}\tilde{u}\left(t\right) \end{array}$$
for which one can easily check that with the new control variable $\tilde{u}$ the orthogonality condition $C{\left(i\right)}^{⊤}D\left(i\right)=0$ holds and $D{\left(i\right)}^{⊤}D\left(i\right)=I_{m_{2}}$, i=1,…,d.

For a multi-agent system with *N* agents, the indexes k,ℓ=1,…,N will be used to define the connection between the agents *k* and *ℓ*.

Some known definitions and results used in the following developments will be briefly reminded (more details and proofs may be found, for instance, in [32,33]).

**Definition 1.** 
*The stochastic system with Markov parameters*

(2)
$$\begin{array}{c} \dot{x}\left(t\right)=A\left(\eta \left(t\right)\right)x\left(t\right) \end{array}$$

*is called exponentially stable in mean square (ESMS) if there exists β≥1 and α>0 such that $E\left[{\left|\Phi \left(t\right)\right|}^{2}|n\left(0\right)=i\right]\leq \beta e^{-\alpha t},\;\forall t\geq 0,\;i\in D$, where Φ(t) denotes the fundamental (random) solution of the differential system (Equation 2).*


**Proposition 1.** 
*The stochastic system (Equation 2) is ESMS if and only if there exist the matrices X(i)>0, i=1,…,d verifying the system of Lyapunov-type inequalities*

(3)
$$\begin{array}{c} A^{⊤}\left(i\right)X\left(i\right)+X\left(i\right)A\left(i\right)+\sum_{j=1}^{d}\pi_{ij}X\left(j\right)<0. \end{array}$$



Throughout the paper it is assumed that the system
(4)$$\begin{array}{c} \begin{array}{ccc} \dot{x}\left(t\right) & = & A\left(\eta (t)\right)x\left(t\right)\\ y(t) & = & C\left(\eta (t)\right)x\left(t\right) \end{array} \end{array}$$
is stochastically detectable, namely there exist a set of matrices H(i),i=1,...,d such that the system $\dot{x}\left(t\right)=\left(A\left(\eta (t)\right)+H\left(\eta (t)\right)C\left(\eta (t)\right)\right)x\left(t\right)$ is ESMS.

The proof of the next result may be found, for instance, in [33] (Theorem 7 of Chapter 3).

**Proposition 2.** 
*If the system (Equation 4) is stochastically detectable and if the system of Lyapunov-type equations*

$$\begin{array}{c} A^{⊤}\left(i\right)X\left(i\right)+X\left(i\right)A\left(i\right)+C^{⊤}\left(i\right)C\left(i\right)+\sum_{j=1}^{d}\pi_{ij}X\left(j\right)=0 \end{array}$$

*has a symmetric solution with X(i)≥0,∀i∈D, then it is ESMS.*


**Proposition 3.** 
*If $v:R^{n}\times D\to R$ is a function of $C^{1}$ class for every i∈D then*

$$\begin{array}{c} \begin{array}{c} E\left[v(x(t),\eta (t))|\eta (0)=i\right]-v\left(x_{0},i\right)=\\ E\left[\int_{0}^{t}\left\{x^{⊤}\left(\tau \right)A^{⊤}\left(\eta \left(\tau \right)\right)+\sum_{j=1}^{d}v\left(x\left(\tau \right),j\right)\pi_{\eta (\tau )j}\right\}d\tau |\eta \left(0\right)=i\right],\;i\in D,\;t\geq 0, \end{array} \end{array}$$

*where x(t) is the solution of the system (Equation 2) with the initial condition $x_{0}$.*


The main result of this section is the following theorem.

**Theorem 1.** 
*If the system of coupled Riccati equations*

(5)
$$\begin{array}{c} \begin{array}{c} A^{⊤}\left(i\right)X\left(i\right)+X\left(i\right)A\left(i\right)+X\left(i\right)\left(\gamma^{-2}B_{1}\left(i\right)B_{1}^{⊤}\left(i\right)-B_{2}\left(i\right)B_{2}^{⊤}\left(i\right)\right)X\left(i\right)\\ +\sum_{j=1}^{d}\pi_{ij}X\left(j\right)+C^{⊤}\left(i\right)C\left(i\right)=0 \end{array} \end{array}$$

*has a stabilizing solution $\left(X(1),...,X(d)\right)$ with X(i)≥0,∀i∈D for a certain γ>0, namely if the stochastic system with Markov jumps*

$$\begin{array}{c} \dot{x}\left(t\right)=\left(A\left(\eta \left(t\right)\right)+\left(\gamma^{-2}B_{1}\left(\eta \left(t\right)\right)B_{1}^{⊤}\left(\eta \left(t\right)\right)-B_{2}\left(\eta \left(t\right)\right)B_{2}^{⊤}\left(\eta \left(t\right)\right)\right)X\left(\eta \left(t\right)\right))\right)x\left(t\right) \end{array}$$

*is ESMS, where*

(6)
$$\begin{array}{c} F\left(\eta \left(t\right)\right):=-B_{2}^{⊤}\left(\eta \left(t\right)\right)X\left(\eta \left(t\right)\right), \end{array}$$

*then the state-feedback control law u(t)=F(η(t))x(t) stabilises the system (Equation 1) and*

(7)
$$\begin{array}{c} E\left[\int_{0}^{\infty }\left(|y_{1}{\left(t\right)|}^{2}-\gamma^{2}{\left|w\left(t\right)\right|}^{2}\right)dt\right]\leq 0 \end{array}$$

*for all $w\in L_{\eta }^{2}\left(\left[0,\infty \right),R^{m_{1}}\right)$, where the quality output $y_{1}\left(t\right)$ is determined with the initial condition x(0)=0 of the system (Equation 1).*


**Proof.** In order to prove that the state feedback gain (Equation 6) stabilises (Equation 1) one may firstly rewrite the Riccati system (Equation 5) as
$$\begin{array}{c} \begin{array}{c} {\left(A\left(i\right)+B_{2}\left(i\right)F\left(i\right)\right)}^{⊤}X\left(i\right)+X\left(i\right)\left(A\left(i\right)+B_{2}\left(i\right)F\left(i\right)\right)+\gamma^{-2}X\left(i\right)B_{1}\left(i\right)B_{1}^{⊤}\left(i\right)X\left(i\right)\\ X\left(i\right)B_{2}\left(i\right)B_{2}^{⊤}\left(i\right)X\left(i\right)+\sum_{j=1}^{d}\pi_{ij}X\left(j\right)+C^{⊤}\left(i\right)C\left(i\right)=0 \end{array} \end{array}$$
with X(i)≥0,i=1,…,d. On the other hand, since the system (Equation 4) is assumed stochastically detectable, it follows that the system
$$\begin{array}{c} \begin{array}{ccc} \dot{x}\left(t\right) & = & \left(A\left(\eta \left(t\right)\right)+B_{2}\left(\eta \left(t\right)\right)F\left(\eta \left(t\right)\right)\right)x\left(t\right)\\ y(t) & = & \left[\begin{array}{c} \gamma^{-1}B_{1}^{⊤}\left(\eta \left(t\right)\right)X\left(\eta \left(t\right)\right)\\ B_{2}^{⊤}\left(\eta \left(t\right)\right)X\left(\eta \left(t\right)\right)\\ C(\eta (t)) \end{array}\right]x\left(t\right) \end{array} \end{array}$$
is also stochastically detectable and then, based on Proposition 2, one concludes that the above stochastic system with the state matrix $A\left(\eta \left(t\right)\right)+B_{2}\left(\eta \left(t\right)\right)F\left(\eta \left(t\right)\right)$ is ESMS. Since the system (Equation 1) with the control law u(t)=F(η(t))x(t) has the same state matrix it follows that the state-feedback (Equation 6) is stabilising.In order to prove the last part of the theorem one introduces the function $V\left(x\left(t\right),\eta \left(t\right)\right)=x^{⊤}\left(t\right)X\left(\eta \left(t\right)\right)x\left(t\right)$. Using Proposition 3 for a certain initial condition $x_{0}$, it follows that
$$\begin{array}{c} \begin{array}{c} E\left[\left(x^{⊤}\left(t\right)X\left(\eta \left(t\right)\right)x\left(t\right)-x_{0}^{⊤}X\left(\eta \left(0\right)\right)x_{0}\right)|\eta \left(0\right)=i\right]\\ =E\left[\int_{0}^{⊤}\left\{2{\left(A\left(\eta \left(\tau \right)\right)x\left(\tau \right)+B_{1}\left(\eta \left(\tau \right)\right)w\left(\tau \right)+B_{2}\left(\eta \left(\tau \right)\right)u\left(\tau \right)\right)}^{⊤}X\left(\eta \left(\tau \right)\right)x\left(\tau \right)\right.\right.\\ \left.\left.+\sum_{j=1}^{d}\pi_{\eta (\tau )j}x^{⊤}\left(\tau \right)X\left(j\right)x\left(\tau \right)\right\}d\tau |\eta \left(0\right)=i\right]. \end{array} \end{array}$$Adding $J\left(i,w,u\right):=E\left[\int_{0}^{t}\left(|y_{1}{\left(\tau \right)|}^{2}-\gamma^{2}{\left|w\left(\tau \right)\right|}^{2}\right)d\tau |\eta \left(0\right)=i\right],\;i\in D$ and using (Equation 5) one obtains
$$\begin{array}{c} \begin{array}{c} J\left(i,w,u\right)+E\left[\left(x^{⊤}\left(t\right)X\left(\eta \left(t\right)\right)x\left(t\right)-x_{0}^{⊤}X\left(\eta \left(0\right)\right)x_{0}\right)|\eta \left(0\right)=i\right]\\ =E\left[\int_{0}^{t}\left\{x^{⊤}\left(\tau \right)C^{⊤}\left(\eta \left(\tau \right)\right)C\left(\eta \left(\tau \right)\right)x\left(\tau \right)+u^{⊤}\left(\tau \right)u\left(\tau \right)-\gamma^{2}w^{⊤}\left(\tau \right)w\left(\tau \right)\right.\right.\\ \left.\left.+x^{⊤}\left(\tau \right)\left(A^{⊤}(\left(\eta \left(\tau \right)\right)X\left(\eta \left(\tau \right)\right)+X\left(\eta \left(\tau \right)\right)A\left(\eta \left(\tau \right)\right)\right)x\left(\tau \right)+w^{⊤}\left(\tau \right)B_{1}^{⊤}\left(\eta \left(\tau \right)\right)X\left(\eta \left(\tau \right)\right)x\left(\tau \right)\right.\right.\\ \left.\left.+x^{⊤}\left(\tau \right)X\left(\eta \left(\tau \right)\right)B_{1}\left(\eta \left(\tau \right)\right)w\left(\tau \right)+u^{⊤}\left(\tau \right)B_{2}^{⊤}\left(\eta \left(\tau \right)\right)X\left(\eta \left(\tau \right)\right)x\left(\tau \right)\right.\right.\\ \left.\left.+x^{⊤}\left(\tau \right)X\left(\eta \left(\tau \right)\right)B_{2}\left(\eta \left(\tau \right)\right)u\left(\tau \right)+\sum_{j=1}^{d}\pi_{\eta (\tau )j}x^{⊤}\left(\tau \right)X\left(j\right)x\left(\tau \right)\right\}d\tau |\eta \left(0\right)=i\right]\\ =E\left[\int_{0}^{t}\left\{x^{⊤}\left(\tau \right)\left(A^{⊤}\left(\eta \left(\tau \right)\right)X\left(\eta \left(\tau \right)\right)+X\left(\eta \left(\tau \right)\right)A\left(\eta \left(\tau \right)\right)\right.\right.\right.\\ \left.\left.\left.+X\left(\eta \left(\tau \right)\right)\left(\gamma^{-2}B_{1}\left(\eta \left(\tau \right)\right)B_{1}^{⊤}\left(\eta \left(\tau \right)\right)-B_{2}\left(\eta \left(\tau \right)\right)B_{2}^{\tau }\left(\eta \left(\tau \right)\right)\right)X\left(\eta \left(\tau \right)\right)\right.\right.\right.\\ \left.\left.\left.+\sum_{j=1}^{d}\pi_{\eta (\tau )j}x^{⊤}\left(\tau \right)X\left(j\right)+C^{⊤}\left(\eta \left(\tau \right)\right)C\left(\eta \left(\tau \right)\right)\right)x\left(\tau \right)\right.\right.\\ \left.\left.+{\left(u\left(\tau \right)+B_{2}^{⊤}\left(\eta \left(\tau \right)\right)X\left(\eta \left(\tau \right)\right)x\left(\tau \right)\right)}^{⊤}\left(u\left(\tau \right)+B_{2}^{⊤}\left(\eta \left(\tau \right)\right)X\left(\eta \left(\tau \right)\right)x\left(\tau \right)\right)\right.\right.\\ \left.\left.-\left(\gamma w\left(\tau \right)-\gamma^{-1}B_{1}^{⊤}\left(\eta \left(\tau \right)\right)X\left(\eta \left(\tau \right)\right)x\left(\tau \right)\right){\left(\gamma w\left(\tau \right)-\gamma^{-1}B_{1}^{⊤}\left(\eta \left(\tau \right)\right)X\left(\eta \left(\tau \right)\right)x\left(\tau \right)\right)}^{⊤}\right\}d\tau |\eta \left(0\right)=i\right]. \end{array} \end{array}$$Taking into account (Equation 5), one obtains
$$\begin{array}{c} \begin{array}{c} J\left(i,w,u\right)+E\left[\left(x^{⊤}\left(t\right)X\left(\eta \left(t\right)\right)x\left(t\right)-x_{0}^{⊤}X\left(\eta \left(0\right)\right)x_{0}\right)|\eta \left(0\right)=i\right]\\ =E\left[\int_{0}^{t}\left\{{\left(u\left(\tau \right)+B_{2}^{⊤}\left(\eta \left(\tau \right)\right)X\left(\eta \left(\tau \right)\right)x\left(\tau \right)\right)}^{⊤}\left(u\left(\tau \right)+B_{2}^{⊤}\left(\eta \left(\tau \right)\right)X\left(\eta \left(\tau \right)\right)x\left(\tau \right)\right)\right.\right.\\ \left.\left.-\left(\gamma w\left(\tau \right)-\gamma^{-1}B_{1}^{⊤}\left(\eta \left(\tau \right)\right)X\left(\eta \left(\tau \right)\right)x\left(\tau \right)\right){\left(\gamma w\left(\tau \right)-\gamma^{-1}B_{1}^{⊤}\left(\eta \left(\tau \right)\right)X\left(\eta \left(\tau \right)\right)x\left(\tau \right)\right)}^{⊤}\right\}d\tau |\eta \left(0\right)=i\right]. \end{array} \end{array}$$For t→∞ and u(t)=F(η(t))x(t), from the above equation it follows that
$$\begin{array}{c} \begin{array}{c} E\left[\int_{0}^{\infty }\left(|y_{1}{\left(t\right)|}^{2}-\gamma^{2}{\left|w\left(t\right)\right|}^{2}\right)dt\right]=\sum_{i=1}^{d}p_{i}\left(0\right)x_{0}^{⊤}X\left(i\right)x_{0}\\ -\sum_{i=1}^{d}p_{i}\left(0\right)E\left[\int_{0}^{\infty }\left(\gamma w\left(\tau \right)-\gamma^{-1}B_{1}^{⊤}\left(\eta \left(\tau \right)\right)X\left(\eta \left(\tau \right)\right)x\left(\tau \right)\right)\right.\\ \left.\times {\left(\gamma w\left(\tau \right)-\gamma^{-1}B_{1}^{⊤}\left(\eta \left(\tau \right)\right)X\left(\eta \left(\tau \right)\right)x\left(\tau \right)\right)}^{⊤}d\tau |\eta \left(0\right)=i\right] \end{array} \end{array}$$
where $p_{i}\left(0\right):=P\left(\eta (0)=i\right)$. Then, for $x_{0}=0$, the inequality (Equation 7) directly follows and it becomes equality for $w^{*}\left(t\right)=\gamma^{-2}B_{1}^{T}X\left(\eta \left(t\right)\right)x\left(t\right)$. □

**Remark 1.** 
*Matrix Riccati equations with indefinite sign as in the system (Equation 5) appear in $H_{\infty }$ control ([34]) and in mixed $H_{2}/H_{\infty }$ control problems ([35]) in the deterministic framework.*


The next result proved in [36] gives a numerical procedure to compute the stabilising solution $\left(X(1),\ldots ,X(d)\right)$ of the system of game theoretic Riccati-type Equation (Equation 5) assuming that such a solution exists.

**Proposition 4.** 
*Assume that the system (Equation 1) is stochastically detectable and that the system of Riccati Equation (Equation 5) has a stabilising solution. Then the sequences ${\left\{X_{k}\left(i\right)\right\}}_{k\geq 0},\;{\left\{Z_{k}\left(i\right)\right\}}_{k\geq 0}$ defined by $X_{0}\left(i\right)=0$ and $X_{k+1}\left(i\right)=X_{k}\left(i\right)+Z_{k}\left(i\right)$, i=1,…,d, where $Z_{0}\left(i\right)$ are the stabilising solutions of the system of Riccati type equations*

(8)
$$\begin{array}{c} \begin{array}{c} {\left(A\left(i\right)+\frac{1}{2}\pi_{ii}I_{n}\right)}^{⊤}Z_{0}\left(i\right)+Z_{0}\left(i\right)\left(A\left(i\right)+\frac{1}{2}\pi_{ii}I_{n}\right)\\ -Z_{0}\left(i\right)B_{2}\left(i\right)B_{2}^{⊤}\left(i\right)Z_{0}\left(i\right)+C^{⊤}\left(i\right)C\left(i\right)+\sum_{j=1,j\neq i}^{d}Z_{0}\left(i\right)=0, \end{array} \end{array}$$

*$Z_{k}\left(i\right),\;k\geq 1$ are the stabilising solutions of the un-coupled Riccati equations*

$$\begin{array}{c} M_{k}^{⊤}\left(i\right)Z_{k}\left(i\right)+Z_{k}\left(i\right)M_{k}\left(i\right)-Z_{k}\left(i\right)B_{2}\left(i\right)B_{2}^{⊤}\left(i\right)Z_{k}\left(i\right)+R_{k}\left(i\right)=0, \end{array}$$

*and where*

$$\begin{array}{c} \begin{array}{ccc} M_{k}\left(i\right) & = & A\left(i\right)+\frac{1}{2}\pi_{ii}I_{n}+\left(\gamma^{-2}B_{1}\left(i\right)B_{1}^{⊤}\left(i\right)-B_{2}\left(i\right)B_{2}^{⊤}\left(i\right)\right)X_{k}\left(i\right)\\ R_{k}\left(i\right) & = & \gamma^{-2}Z_{k-1}\left(i\right)B_{1}\left(i\right)B_{1}^{⊤}\left(i\right)Z_{k-1}\left(i\right)+\sum_{j=1,j\neq i}^{d}\pi_{ij}Z_{k-1}\left(j\right), \end{array} \end{array}$$

*are convergent and the limit of $X_{k}\left(i\right),\;i=1,\ldots ,d$ when k→∞ is the stabilising solution of the system (Equation 5).*


An iterative algorithm to solve the system of coupled Riccati equations with definite sign is given for completeness in Appendix A. The proof of the algorithm convergence may be found in [33] (Theorem 21 of Chapter 4).

## 3. Markovian $H_{\infty }$ Controller Design for Multi-Agent Systems

Consider N>1 agents with identical dynamics of form
(9)$$\begin{array}{c} \begin{array}{ccc} {\dot{x}}_{k}\left(t\right) & = & A\left(\eta \left(t\right)\right)x_{k}\left(t\right)+B_{1}\left(\eta \left(t\right)\right)w_{k}\left(t\right)+B_{2}\left(\eta \left(t\right)\right)u_{k}\left(t\right)\\ y_{1k}\left(t\right) & = & C\left(\eta \left(t\right)\right)x_{k}\left(t\right)+D\left(\eta \left(t\right)\right)u_{k}\left(t\right)\\ y_{2k}\left(t\right) & = & x_{k}\left(t\right),\;t\geq 0,k=1,\ldots ,N \end{array} \end{array}$$
with $C^{⊤}\left(i\right)D\left(i\right)=0$ and $D^{⊤}\left(i\right)D\left(i\right)=I_{m_{1}}$, i=1,…,d, and k=1,…,N.

**Remark 2.** 
*Although in (Equation 9) one considered the same standard homogeneous Markov chain for all agents one may also treat the case when each agent is modelled with its own stochastic process $\eta_{k}\left(t\right)$. Indeed, if each agent dynamics is modelled with a standard homogenous Markov chain $\eta_{k}\left(t\right),\;k=1,...,N$ with d states then one may consider for the multi-agent system (Equation 9) an extended Markov chain with $d^{N}$ states. Since in the present paper the Markov parameters are used to characterise the availability or the link failure between agents it follows that the maximal number of states of the Markov chain in these applications is $2^{N}$.*


The dynamics of the multi-agent system (Equation 9) may be written in the following compact form
(10)$$\begin{array}{c} \begin{array}{ccc} \dot{\tilde{x}}\left(t\right) & = & \tilde{A}\left(\eta \left(t\right)\right)\tilde{x}\left(t\right)+{\tilde{B}}_{1}\left(\eta \left(t\right)\right)\tilde{w}\left(t\right)+{\tilde{B}}_{2}\left(\eta \left(t\right)\right)\tilde{u}\left(t\right)\\ {\tilde{y}}_{1}\left(t\right) & = & \tilde{C}\left(\eta \left(t\right)\right)\tilde{x}\left(t\right)+\tilde{D}\left(\eta \left(t\right)\right)\tilde{u}\left(t\right)\\ {\tilde{y}}_{2}\left(t\right) & = & \tilde{x}\left(t\right) \end{array} \end{array}$$
in which $\tilde{x}:={\left[x_{1}^{⊤},\ldots ,x_{N}^{⊤}\right]}^{⊤}$, $\tilde{w}:={\left[w_{1}^{⊤},\ldots ,w_{N}^{⊤}\right]}^{⊤}$, $\tilde{u}:=[u_{1}^{⊤},\ldots ,$$u_{N}^{⊤}{]}^{⊤}$, ${\tilde{y}}_{1}:={\left[y_{11}^{⊤},\ldots ,y_{1N}^{⊤}\right]}^{⊤}$ and $\tilde{A}\left(\eta \left(t\right)\right):=I_{N}⊗A\left(\eta \left(t\right)\right)$, ${\tilde{B}}_{1}\left(\eta \left(t\right)\right):=I_{N}⊗B_{1}\left(\eta \left(t\right)\right)$, ${\tilde{B}}_{2}\left(\eta \left(t\right)\right):=I_{N}⊗B_{2}\left(\eta \left(t\right)\right)$, $\tilde{C}\left(\eta \left(t\right)\right):=I_{N}⊗C\left(\eta \left(t\right)\right)$, $\tilde{D}\left(\eta \left(t\right)\right):=I_{N}⊗D\left(\eta \left(t\right)\right)$ where ⊗ denotes the Kronecker product.

For γ>0 define the cost function
(11)$$\begin{array}{c} \begin{array}{c} J\left(\tilde{w},\;\tilde{u}\right)=E\left[\int_{0}^{\infty }\left\{|{\tilde{y}}_{1}{\left(t\right)|}^{2}-\gamma^{2}{\left|\tilde{w}\left(t\right)\right|}^{2}\right.\right.\\ \left.\left.+\frac{1}{2}\sum_{k=1}^{N}\sum_{\ell =1,\ell \neq k}^{N}{\left(x_{k}\left(t\right)-x_{\ell }\left(t\right)\right)}^{⊤}Q_{k\ell }\left(\eta \left(t\right)\right)\left(x_{k}\left(t\right)-x_{\ell }\left(t\right)\right)\right\}dt\right] \end{array} \end{array}$$
where $Q_{k\ell }\left(i\right),\;k,\ell =1,\ldots ,N$ and i=1,…,d are positive semidefinite weighting matrices. Then (Equation 11) may be rewritten as
(12)$$\begin{array}{c} J\left(\tilde{w},\;\tilde{u}\right)=E\left[\int_{0}^{\infty }\left({\tilde{y}}_{1}{\left(t\right)}^{⊤}{\tilde{y}}_{1}\left(t\right)-\gamma^{2}\tilde{w}{\left(t\right)}^{⊤}\tilde{w}\left(t\right)+\tilde{x}{\left(t\right)}^{⊤}\tilde{Q}\left(\eta \left(t\right)\right)\tilde{x}\left(t\right)\right)dt\right] \end{array}$$
in which $\tilde{Q}\left(i\right)$ has the block elements
(13)$$\begin{array}{c} \begin{array}{ccc} {\tilde{Q}}_{kk}\left(i\right) & = & C^{⊤}\left(i\right)C\left(i\right)+\sum_{\ell =1,\ell \neq k}^{N}Q_{k\ell }\left(i\right),\\ {\tilde{Q}}_{k\ell }\left(i\right) & = & -Q_{k\ell }\left(i\right),\;k,\ell =1,\ldots ,N,\;k\neq \ell ,\; \end{array} \end{array}$$
i=1,…,d. Choosing $Q_{k\ell }\left(i\right)=P^{⊤}\left(i\right)P\left(i\right),\;k,\ell =1,\ldots ,N,\;k\neq \ell$, i=1,…,d, it follows that the block elements of the matrix $\tilde{Q}\left(i\right)$ are
$$\begin{array}{c} \begin{array}{ccc} {\tilde{Q}}_{kk}\left(i\right) & = & C^{⊤}\left(i\right)C\left(i\right)+\left(N-1\right)P^{⊤}\left(i\right)P\left(i\right),\\ {\tilde{Q}}_{k\ell }\left(i\right) & = & -P^{⊤}\left(i\right)P\left(i\right),\;k,\ell =1,\ldots ,N,\;k\neq \ell ,\; \end{array} \end{array}$$
i=1,…,d. One can easily check that
$$\begin{array}{c} {\tilde{y}}_{1}{\left(t\right)}^{⊤}{\tilde{y}}_{1}\left(t\right)+\tilde{x}\left(t\right)\tilde{Q}\left(\eta \left(t\right)\right)\tilde{x}\left(t\right)=\tilde{C}\left(\eta \left(t\right)\right)\tilde{x}\left(t\right)+\tilde{D}\left(\eta \left(t\right)\right)\tilde{u}\left(t\right) \end{array}$$
where
(14)$$\tilde{C}\left(i\right):=\left[\begin{array}{cccc} & & \tilde{P}\left(i\right)\;\; & \;\\ C(i) & 0 & \ldots & 0\\ 0 & C(i) & \ldots & 0\\ \vdots & \vdots & \ddots & \vdots \\ 0 & 0 & \ldots & C(i) \end{array}\right]\;,\tilde{D}\left(i\right):=\left[\begin{array}{cccc} \; & \; & 0_{n\cdot N\times m_{2}\cdot N} & \;\\ D(i) & 0 & \ldots & 0\\ 0 & D(i) & \ldots & 0\\ \vdots & \vdots & \ddots & \vdots \\ 0 & 0 & \ldots & D(i) \end{array}\right],\;$$
i=1,...,d with $\tilde{P}\left(i\right)$ satisfying the condition
$$\begin{array}{c} \tilde{P}{\left(i\right)}^{⊤}\tilde{P}\left(i\right)=\left[\begin{array}{cccc} \left(N-1\right)P^{⊤}\left(i\right)P\left(i\right) & -P^{⊤}\left(i\right)P\left(i\right) & \ldots & -P^{⊤}\left(i\right)P\left(i\right)\\ -P^{⊤}\left(i\right)P\left(i\right) & \left(N-1\right)P^{⊤}\left(i\right)P\left(i\right) & \ldots & -P^{⊤}\left(i\right)P\left(i\right)\\ \vdots & \vdots & \ddots & \vdots \\ -P^{⊤}\left(i\right)P\left(i\right) & -P^{⊤}\left(i\right)P\left(i\right) & \ldots & \left(N-1\right)P^{⊤}\left(i\right)P\left(i\right) \end{array}\right], \end{array}$$
i=1,…,d. Therefore the cost function (Equation 11) may be rewritten as
(15)$$\begin{array}{c} J\left(\tilde{w},\tilde{u}\right)=E\left[\int_{0}^{\infty }\left({\tilde{z}}^{⊤}\left(t\right)\tilde{z}\left(t\right)-\gamma^{2}{\tilde{w}}^{⊤}\left(t\right)\tilde{w}\left(t\right)\right)dt\right] \end{array}$$
where $z\left(t\right)=\tilde{C}\left(\eta \left(t\right)\right)\tilde{x}\left(t\right)+\tilde{D}\left(\eta \left(t\right)\right)\tilde{u}\left(t\right)$. Moreover, since it was assumed that $C^{⊤}\left(i\right)D\left(i\right)=0$ and $D^{⊤}\left(i\right)D\left(i\right)=I_{m_{2}},\;i=1,\ldots ,d$ it follows that ${\tilde{C}}^{⊤}\left(i\right)\tilde{D}\left(i\right)=0$ and ${\tilde{D}}^{⊤}\left(i\right)\tilde{D}\left(i\right)=I_{m_{1}\cdot N},\;i=1,\ldots ,d$ and therefore one may apply Theorem 1 for the Markov stochastic system
(16)$$\begin{array}{c} \begin{array}{ccc} \dot{\tilde{x}}\left(t\right) & = & \tilde{A}\left(\eta \left(t\right)\right)x\left(t\right)+{\tilde{B}}_{1}\left(\eta \left(t\right)\right)w\left(t\right)+{\tilde{B}}_{2}\left(\eta \left(t\right)\right)u\left(t\right)\\ \tilde{z}\left(t\right) & = & \tilde{C}\left(\eta \left(t\right)\right)\tilde{x}\left(t\right)+\tilde{D}\left(\eta \left(t\right)\right)\tilde{u}\left(t\right)\\ {\tilde{y}}_{2}\left(t\right) & = & \tilde{x}\left(t\right) \end{array} \end{array}$$
with the cost function (Equation 15).

The main result of this section is given by the following theorem.

**Theorem 2.** 
*(i) If the system of coupled Riccati equations*

(17)
$$\begin{array}{c} \begin{array}{c} {\tilde{A}}^{⊤}\left(i\right)\tilde{X}\left(i\right)+\tilde{X}\left(i\right)\tilde{A}\left(i\right)+\tilde{X}\left(i\right)\left(\gamma^{-2}{\tilde{B}}_{1}\left(i\right){\tilde{B}}_{1}^{⊤}\left(i\right)-{\tilde{B}}_{2}\left(i\right){\tilde{B}}_{2}^{⊤}\left(i\right)\right)\tilde{X}\left(i\right)\\ +\sum_{j=1}^{d}\pi_{ij}\tilde{X}\left(j\right)+{\tilde{Q}}^{⊤}\left(i\right)\tilde{Q}\left(i\right)=0,\;\;\;i=1,\ldots ,d. \end{array} \end{array}$$

*has a stabilising solution $\left(\tilde{X}\left(1\right),\ldots ,\tilde{X}\left(d\right)\right)$ with $\tilde{X}\left(i\right)\geq 0,\;i=1,\ldots ,d$ then the stochastic system with Markov parameters*

(18)
$$\begin{array}{c} \dot{\tilde{x}}\left(t\right)=\left(\tilde{A}\left(\eta \left(t\right)\right)+{\tilde{B}}_{2}\left(\eta \left(t\right)\right)\tilde{F}\left(\eta \left(t\right)\right)\right)x\left(t\right)+{\tilde{B}}_{1}\left(\eta \left(t\right)\right)\tilde{w}\left(t\right) \end{array}$$

*where $\tilde{F}\left(i\right)=-{\tilde{B}}_{2}^{⊤}\left(i\right)\tilde{X}\left(i\right)$, i=1,…,d, is ESMS and for the initial condition $\tilde{x}\left(0\right)=0$,*

$$\begin{array}{c} E\left[\int_{0}^{\infty }\left(|{\tilde{y}}_{1}{\left(t\right)|}^{2}-\gamma^{2}{\left|\tilde{w}\left(t\right)\right|}^{2}\right)dt\right]\leq 0 \end{array}$$

*for all $\tilde{w}\in L_{\eta }^{2}\left(\left[0,\infty \right),R^{N\cdot m_{1}}\right)$.*

*(ii) The solution of (Equation 17) has the following structure*

(19)
$$\begin{array}{c} \begin{array}{ccc} \tilde{X}\left(i\right) & := & {\left[{\tilde{X}}_{k\ell }\right]}_{k,\ell =1,\ldots ,N}\;\;where\\ {\tilde{X}}_{kk}\left(i\right) & := & X_{1}\left(i\right)+\left(N-1\right)X_{2}\left(i\right)\\ {\tilde{X}}_{k\ell }\left(i\right) & := & -X_{2}\left(i\right),\;k,\;\ell =1,\ldots ,N,\;\;k\neq \ell , \end{array} \end{array}$$

*in which $\left(X_{1}\left(1\right),\ldots ,X_{1}\left(d\right)\right)$ and $\left(X_{2}\left(1\right),\ldots ,X_{2}\left(d\right)\right)$ are the solutions of the Riccati type equations*

(20)
$$\begin{array}{c} \begin{array}{c} A^{⊤}\left(i\right)X_{1}\left(i\right)+X_{1}\left(i\right)A\left(i\right)+X_{1}\left(i\right)\left(\gamma^{-2}B_{1}\left(i\right)B_{1}^{⊤}\left(i\right)-B_{2}\left(i\right)B_{2}^{⊤}\left(i\right)\right)X_{1}\left(i\right)\\ +\sum_{j=1}^{d}\pi_{ij}X_{1}\left(j\right)+C^{⊤}\left(i\right)C\left(i\right)=0,\;\;i=1,\ldots ,d \end{array} \end{array}$$

*and*

(21)
$$\begin{array}{c} \begin{array}{c} {\left[A\left(i\right)+\left(\gamma^{-2}B_{1}\left(i\right)B_{1}^{⊤}\left(i\right)-B_{2}\left(i\right)B_{2}^{⊤}\left(i\right)\right)X_{1}\left(i\right)\right]}^{⊤}X_{2}\left(i\right)\\ +X_{2}\left(i\right)\left[(A\left(i\right)+\left(\gamma^{-2}B_{1}\left(i\right)B_{1}^{⊤}\left(i\right)-B_{2}\left(i\right)B_{2}^{⊤}\left(i\right)\right)X_{1}\left(i\right)\right]\\ +NX_{2}\left(i\right)\left(\gamma^{-2}B_{1}\left(i\right)B_{1}^{⊤}\left(i\right)-B_{2}\left(i\right)B_{2}^{⊤}\left(i\right)\right)X_{2}\left(i\right)\\ +\sum_{j=1}^{d}\pi_{ij}X_{2}\left(j\right)+P^{⊤}\left(i\right)P\left(i\right)=0,\;i=1,\ldots ,d, \end{array} \end{array}$$

*respectively.*

*(iii) If the Riccati type systems (Equation 20) and (Equation 21) have the stabilising solutions $\left(X_{1}\left(1\right),\ldots ,X_{1}\left(d\right)\right)$ and $\left(X_{2}\left(1\right),\ldots ,X_{2}\left(d\right)\right)$ respectively, with $X_{1}\left(i\right)\geq 0$ and $X_{2}\left(i\right)\geq 0$, i=1,…,d then $\left(\tilde{X}\left(1\right),...,\tilde{X}\left(d\right)\right)$ with $\tilde{X}\left(i\right)$ defined in (Equation 19), is the stabilising solution of (Equation 17) and $\tilde{X}\left(i\right)\geq 0,\;i=1,\ldots ,d$.*


**Proof.** Part (i) of the statement is a direct consequence of Theorem 1.The proof of (ii) is inspired by the arguments given in Theorems 1 and 2 of [26] for the multi agent linear quadratic control problem in deterministic framework. Thus, the stabilising solution $\left(\tilde{X}\left(1\right),\ldots ,\tilde{X}\left(d\right)\right)$ of (Equation 17) has an N×N blocks structure, each of them having the size (n×n). Since all the matrix coefficients in (Equation 17) are diagonal it follows that the diagonal and the off-diagonal elements of $\tilde{X}\left(i\right),\;i=1,\ldots ,d$ are equal, respectively. Then the diagonal block elements will be denoted by ${\tilde{X}}_{1}\left(i\right)$ and the off-diagonal ones by ${\tilde{X}}_{2}\left(i\right),\;i=1,\ldots ,d$. The blocks (k,ℓ),k,ℓ∈{1,…,N} of (Equation 17) have the form
$$\begin{array}{c} \begin{array}{c} A^{⊤}\left(i\right){\tilde{X}}_{k\ell }\left(i\right)+{\tilde{X}}_{k\ell }\left(i\right)A\left(i\right)+\sum_{m=1}^{N}{\tilde{X}}_{km}\left(i\right)\left(\gamma^{-2}B_{1}\left(i\right)B_{1}^{⊤}\left(i\right)-B_{2}\left(i\right)B_{2}^{⊤}\left(i\right)\right){\tilde{X}}_{m\ell }\left(i\right)\\ +{\tilde{Q}}_{kl}\left(i\right)+\sum_{j=1}^{d}\pi_{ij}{\tilde{X}}_{k\ell }\left(j\right)=0. \end{array} \end{array}$$
Denoting $X_{1}\left(i\right):={\tilde{X}}_{1}\left(i\right)+\left(N-1\right){\tilde{X}}_{2}\left(i\right),\;i=1,\ldots ,d$ and summing up the terms of the *k*-th row of (Equation 17) it follows that $X_{1}\left(i\right),\;i=1,\ldots ,d$ verifies (Equation 20). Further, for any off-diagonal block (k,ℓ) with k≠ℓ and k,ℓ∈{1,…,N}, direct computations give
$$\begin{array}{c} \begin{array}{c} {\left(A\left(i\right)+\left(\gamma^{-2}B_{1}\left(i\right)B_{1}^{⊤}\left(i\right)-B_{2}\left(i\right)B_{2}^{⊤}\left(i\right)\right){\tilde{X}}_{1}\left(i\right)\right)}^{⊤}{\tilde{X}}_{2}\left(i\right)\\ +{\tilde{X}}_{2}\left(i\right)\left(A\left(i\right)+\left(\gamma^{-2}B_{1}\left(i\right)B_{1}^{⊤}\left(i\right)-B_{2}\left(i\right)B_{2}^{⊤}\left(i\right)\right){\tilde{X}}_{1}\left(i\right)\right)\\ +\left(N-2\right){\tilde{X}}_{2}\left(i\right)\left(\gamma^{-2}B_{1}\left(i\right)B_{1}^{⊤}\left(i\right)-B_{2}\left(i\right)B_{2}^{⊤}\left(i\right)\right){\tilde{X}}_{2}\left(i\right)\\ +{\tilde{Q}}_{k\ell }\left(i\right)+\sum_{j=1}^{d}\pi_{ij}{\tilde{X}}_{2}\left(j\right)=0. \end{array} \end{array}$$
Using the fact that ${\tilde{X}}_{1}\left(i\right)=X_{1}\left(i\right)-\left(N-1\right){\tilde{X}}_{2}\left(i\right)$ and that ${\tilde{Q}}_{k\ell }\left(i\right)=-P^{⊤}\left(i\right)P\left(i\right),\;i=1,\ldots ,d$ from the above equation it follows that $X_{2}\left(i\right):=-{\tilde{X}}_{2}\left(i\right)$ is a solution of (Equation 21).For part (iii) of the statement, one will prove that the stochastic system
(22)$$\begin{array}{c} \dot{\tilde{x}}\left(t\right)=\left(\tilde{A}\left(\eta \left(t\right)\right)+\left(\gamma^{-2}{\tilde{B}}_{1}\left(\eta \left(t\right)\right){\tilde{B}}_{1}^{⊤}\left(\eta \left(t\right)\right)-{\tilde{B}}_{2}\left(\eta \left(t\right)\right){\tilde{B}}_{2}^{⊤}\left(\eta \left(t\right)\right)\right)\tilde{X}\left(\eta \left(t\right)\right)\right)\tilde{x}\left(t\right) \end{array}$$
is ESMS, namely there exist the positive definite matrices $\tilde{S}\left(i\right),\;i=1,\ldots ,d$ such that
(23)$$\begin{array}{c} \begin{array}{c} {\left(\tilde{A}\left(\eta \left(t\right)\right)+\left(\gamma^{-2}{\tilde{B}}_{1}\left(\eta \left(t\right)\right){\tilde{B}}_{1}^{⊤}\left(\eta \left(t\right)\right)-{\tilde{B}}_{2}\left(\eta \left(t\right)\right){\tilde{B}}_{2}^{⊤}\left(\eta \left(t\right)\right)\right)\tilde{X}\left(\eta \left(t\right)\right)\right)}^{⊤}\tilde{S}\left(i\right)\\ +\tilde{S}\left(i\right)\left(\tilde{A}\left(\eta \left(t\right)\right)+\left(\gamma^{-2}{\tilde{B}}_{1}\left(\eta \left(t\right)\right){\tilde{B}}_{1}^{⊤}\left(\eta \left(t\right)\right)-{\tilde{B}}_{2}\left(\eta \left(t\right)\right){\tilde{B}}_{2}^{⊤}\left(\eta \left(t\right)\right)\right)\tilde{X}\left(\eta \left(t\right)\right)\right)\\ +\sum_{j=1}^{d}\pi_{ij}\tilde{S}\left(j\right)<0,\;i=1,\ldots ,d. \end{array} \end{array}$$
Since $\left(X_{1}\left(1\right),\ldots ,X_{1}\left(d\right)\right)$ is the stabilising solution of the Riccati-type system (Equation 20) it follows that the stochastic system
$$\begin{array}{c} \dot{x}\left(t\right)=\left(A\left(\eta \left(t\right)\right)+M\left(\eta \left(t\right)\right)X_{1}\left(\eta \left(t\right)\right)\right)x\left(t\right), \end{array}$$
where one denoted $M\left(i\right):=\gamma^{-2}B_{1}\left(i\right)B_{1}^{⊤}\left(i\right)-B_{2}\left(i\right)B_{2}^{⊤}\left(i\right)$, is ESMS and therefore there exist the positive definite matrices $S_{1}\left(i\right),i=1,\ldots ,d$ such that
(24)$$\begin{array}{c} {\left(A\left(i\right)+M\left(i\right)X_{1}\left(i\right)\right)}^{⊤}S_{1}\left(i\right)+S_{1}\left(i\right)\left(A\left(i\right)+M\left(i\right)X_{1}\left(i\right)\right)+\sum_{j=1}^{d}\pi_{ij}S_{1}\left(j\right)<0. \end{array}$$
Similarly, based on the fact that $\left(X_{2}\left(1\right),\ldots ,X_{2}\left(d\right)\right)$ is the stabilising solution of (Equation 21), there exist the positive definite matrices $S_{2}\left(i\right),\;i=1,\ldots ,d$ such that
(25)$$\begin{array}{c} \begin{array}{c} {\left(A\left(i\right)+M\left(i\right)\left(X_{1}\left(i\right)+NX_{2}\left(i\right)\right)\right)}^{⊤}S_{2}\left(i\right)\\ +S_{2}\left(i\right)\left(A\left(i\right)+M\left(i\right)\left(X_{1}\left(i\right)+NX_{2}\left(i\right)\right)\right)+\sum_{j=1}^{d}\pi_{ij}S_{2}\left(j\right)<0,\;i=1,\ldots ,d. \end{array} \end{array}$$
Define
$$\begin{array}{c} \tilde{S}\left(i\right)=\left[\begin{array}{cccc} S_{1}\left(i\right)+\left(N-1\right)S_{2}\left(i\right) & -S_{2}\left(i\right) & \ldots & -S_{2}\left(i\right)\\ -S_{2}\left(i\right) & S_{1}\left(i\right)+\left(N-1\right)S_{2}\left(i\right) & \ldots & -S_{2}\left(i\right)\\ \vdots & \vdots & \ddots & \vdots \\ -S_{2}\left(i\right) & -S_{2}\left(i\right) & \ldots & S_{1}\left(i\right)+\left(N-1\right)S_{2}\left(i\right) \end{array}\right] \end{array}$$
for which the inequalities (Equation 23) become:
(26)$$\begin{array}{c} P\left(i\right):=\left[\begin{array}{cccc} P_{1}\left(i\right) & P_{2}\left(i\right) & \ldots & P_{2}\left(i\right)\\ P_{2}\left(i\right) & P_{1}\left(i\right) & \ldots & P_{2}\left(i\right)\\ \vdots & \vdots & \ddots & \vdots \\ P_{2}\left(i\right) & P_{2}\left(i\right) & \ldots & P_{1}\left(i\right) \end{array}\right]<0,\;i=1,\ldots ,d \end{array}$$
where
$$\begin{array}{c} \begin{array}{c} P_{1}\left(i\right)={\left(A\left(i\right)+M\left(i\right)\left(X_{1}\left(i\right)+\left(N-1\right)X_{2}\left(i\right)\right)\right)}^{⊤}\left(S_{1}\left(i\right)+\left(N-1\right)S_{2}\left(i\right)\right)\\ +\left(S_{1}\left(i\right)+\left(N-1\right)S_{2}\left(i\right)\right)\left(A\left(i\right)+M\left(i\right)\left(X_{1}\left(i\right)+\left(N-1\right)X_{2}\left(i\right)\right)\right)\\ +\left(N-1\right)\left(X_{2}\left(i\right)M\left(i\right)S_{2}\left(i\right)+S_{2}\left(i\right)M\left(i\right)X_{2}\left(i\right)\right)+\sum_{j=1}^{d}\pi_{ij}\left(S_{1}\left(j\right)+\left(N-1\right)S_{2}\left(j\right)\right) \end{array} \end{array}$$
and
$$\begin{array}{c} \begin{array}{c} P_{2}\left(i\right)=-{\left(A\left(i\right)+M\left(i\right)\left(X_{1}\left(i\right)+\left(N-1\right)X_{2}\left(i\right)\right)\right)}^{⊤}S_{2}\left(i\right)\\ -S_{2}\left(i\right)\left(A\left(i\right)+M\left(i\right)\left(X_{1}\left(i\right)+\left(N-1\right)X_{2}\left(i\right)\right)\right)\\ -\left(S_{1}\left(i\right)+S_{2}\left(i\right)\right)M\left(i\right)X_{2}\left(i\right)-X_{2}\left(i\right)M\left(i\right)\left(S_{1}\left(i\right)+S_{2}\left(i\right)\right)-\sum_{j=1}^{d}\pi_{ij}S_{2}\left(j\right), \end{array} \end{array}$$
i=1,…,d. Defining the matrix
$$\begin{array}{c} T=\left[\begin{array}{ccccc} I_{n} & I_{n} & I_{n} & \ldots & I_{n}\\ 0 & I_{n} & 0 & \ldots & 0\\ \vdots & \vdots & \ddots & \vdots & \vdots \\ 0 & 0 & 0 & \ldots & I_{n} \end{array}\right] \end{array}$$
direct computations give
$$\begin{array}{c} TP\left(i\right)T^{-1}=\left[\begin{array}{cccc} P_{1}\left(i\right)+\left(N-1\right)P_{2}\left(i\right) & 0 & \ldots & 0\\ ☆ & P_{1}\left(i\right)-P_{2}\left(i\right) & \ldots & 0\\ ☆ & ☆ & \ddots & 0\\ ☆ & ☆ & \ldots & P_{1}\left(i\right)-P_{2}\left(i\right) \end{array}\right] \end{array}$$
in which ∗ denotes irrelevant elements. Due to the triangular structure of the above system it follows that the spectra of P(i),i=1,…,d is given by the reunion of the spectra of $P_{1}\left(i\right)+\left(N-1\right)P_{2}\left(i\right)$ and of $P_{1}\left(i\right)-P_{2}\left(i\right)$, respectively, i=1,…,d. On the other hand, one may direcly check that $P_{1}\left(i\right)+\left(N-1\right)P_{2}\left(i\right)$ coincide with the left hand sides of (Equation 24), i=1,…,d. Further, using the above expressions of $P_{1}\left(i\right)$ and $P_{2}\left(i\right)$, it follows that
(27)$$\begin{array}{c} \begin{array}{c} P_{1}\left(i\right)-P_{2}\left(i\right)={\left(A\left(i\right)+M\left(i\right)\left(X_{1}\left(i\right)+NX_{2}\left(i\right)\right)\right)}^{⊤}\left(S_{1}\left(i\right)+NS_{2}\left(i\right)\right)\\ +\left(S_{1}\left(i\right)+NS_{2}\left(i\right)\right)\left(A\left(i\right)+M\left(i\right)\left(X_{1}\left(i\right)+NX_{2}\left(i\right)\right)\right)+\sum_{j=1}^{d}\pi_{ij}\left(S_{1}\left(i\right)+NS_{2}\left(i\right)\right),\; \end{array} \end{array}$$
i=1,…,d. Since (Equation 24) remains true if $S_{1}\left(i\right)$ are replaced by $\epsilon S_{1}\left(i\right),\;i=1,\ldots ,d$ for any ϵ>0, from (Equation 25) it follows that for a small enough ϵ>0, $P_{1}\left(i\right)-P_{2}\left(i\right)<0$. Thus one concludes that P(i)<0,i=1,…,d and therefore $\left(\tilde{X}\left(1\right),\ldots ,\tilde{X}\left(d\right)\right)$ is the stabilising solution of the Riccati system (Equation 17). The fact that $\tilde{X}\left(i\right),\;i=1,\ldots ,d$ given by (Equation 19) are positive semidefinite directly follows taking into account that
$$\begin{array}{c} T\tilde{X}\left(i\right)T^{-1}=\left[\begin{array}{cccc} X_{1}\left(i\right) & 0 & \ldots & 0\\ ☆ & X_{1}\left(i\right)+NX_{2}\left(i\right) & \ldots & 0\\ ☆ & ☆ & \ddots & 0\\ ☆ & ☆ & \ldots & X_{1}\left(i\right)+NX_{2}\left(i\right) \end{array}\right] \end{array}$$
and using the assumption that $X_{1}\left(i\right)$ and $X_{2}\left(i\right)$ are positive semidefinite. Thus the proof ends. □

**Remark 3.** 
*Based on the expressions of $\tilde{X}\left(i\right)$ and $\tilde{F}\left(i\right),\;i=1,\ldots ,d$, it follows that $\tilde{F}\left(i\right)$ has the following structure*

(28)
$$\begin{array}{c} \tilde{F}\left(i\right)=\left[\begin{array}{cccc} F_{1}\left(i\right) & F_{2}\left(i\right) & \ldots & F_{2}\left(i\right)\\ F_{2}\left(i\right) & F_{1}\left(i\right) & \ldots & F_{2}\left(i\right)\\ \vdots & \ddots & \vdots & \vdots \\ F_{2}\left(i\right) & F_{2}\left(i\right) & \ldots & \;F_{1}\left(i\right) \end{array}\right] \end{array}$$

*where*

(29)
$$\begin{array}{c} \begin{array}{ccc} F_{1}\left(i\right) & = & -B_{2}{\left(i\right)}^{⊤}\left(X_{1}\left(i\right)+\left(N-1\right)X_{2}\left(i\right)\right)\\ F_{2}\left(i\right) & = & B_{2}{\left(i\right)}^{⊤}X_{2}\left(i\right),\;i=1,\ldots ,d. \end{array} \end{array}$$



## 4. The Data Packet Dropout Case

From (Equation 26) and (Equation 29) it follows that the control of each agent is determined as a combination of its own states and the states of all other network agents. The case when the states of all agents are available for every agent represents the *nominal case* and it will be denoted as the state i=1 of the set D. In the case when the state of a certain agent is not available, the corresponding terms in the control expression of the other agents will be zero which is equivalent with the fact that ${\tilde{F}}_{2}\left(2\right)=0$. This case will be associated with the state i=2 of D. Supposing that the communication network fails at a certain moment of time, it follows that $\tilde{F}\left(2\right)=diag\left(F_{1}\left(2\right),\ldots ,F_{1}\left(2\right)\right)$ and $X_{2}\left(2\right)=0$. The condition $X_{2}\left(2\right)=0$ is accomplished if the weights corresponding to the coupling terms in (Equation 11) are taken to be zero, namely if P(2)=0 and if the corresponding row in the stationary transition rate matrix Π has null elements. To conclude, in the above considered scenario in which either the communication network properly works or it completely fails leads to a Markovian model with d=2 states of the set D. If the connection with a single agent, let say *k* is lost then the gain $\tilde{F}$ will have all extra diagonal elements of the *k*-th column equal to zero. Similarly, in the case when the connections with more agents fail, the columns of $\tilde{F}$ corresponding to these agents will have zero extra diagonal elements; in this situation the Markov chain will have maximum $2^{N}$ states.

In order to illustrate these ideas one considers a networked system with N=100 agents whose planar motions are described by the kinematic equations
$$\begin{array}{c} \begin{array}{ccc} {\ddot{x}}_{k}\left(t\right) & = & u_{k}\left(t\right)\\ {\ddot{y}}_{k}\left(t\right) & = & v_{k}\left(t\right),\;k=1,\ldots ,N \end{array} \end{array}$$
where $x_{k}$ and $y_{k}$ denote the Cartesian coordinates and $u_{i}$ and $v_{i}$ their commanded accelerations, respectively. Two states of the Markov chain have been considered in this example. The first one corresponds to the case when the communication properly works and the states of all agents are available for all others; the second state of the Markov chain characterizes the situation in which each agent can only access its own state vector. Assuming that the quality outputs (denoted in Section 3 by $y_{1k}$) are ${\left[\begin{array}{cccc} x_{k} & y_{k} & u_{k} & v_{k} \end{array}\right]}^{⊤}$ it follows that the matrix coefficients in the representation (Equation 1) are identical for both states of the Markov chain, namely
$$\begin{array}{c} A\left(i\right)=\left[\begin{array}{cccc} 0 & 1 & 0 & 0\\ 0 & 0 & 0 & 0\\ 0 & 0 & 0 & 1\\ 0 & 0 & 0 & 0 \end{array}\right],\;B_{1}\left(i\right)=\left[\begin{array}{cc} 1 & 0\\ 0 & 0\\ 0 & 1\\ 0 & 0 \end{array}\right],\;B_{2}\left(i\right)=\left[\begin{array}{cc} 0 & 0\\ 1 & 0\\ 0 & 0\\ 0 & 1 \end{array}\right],\\ C\left(i\right)=\left[\begin{array}{cccc} 1 & 0 & 0 & 0\\ 0 & 0 & 1 & 0\\ 0 & 0 & 0 & 0\\ 0 & 0 & 0 & 0 \end{array}\right],\;D\left(i\right)=\left[\begin{array}{cc} 0 & 0\\ 0 & 0\\ 1 & 0\\ 0 & 1 \end{array}\right],\;i=1,2, \end{array}$$
only the weights $Q_{k\ell }\left(i\right)$ in the cost function (Equation 11) being different for the two states. Thus for i=1 one chose $Q_{k\ell }=100\cdot I_{4}$ (namely $P\left(1\right)=10^{2}\cdot I_{4}$) and for i=2, $Q_{k\ell }\left(2\right)=0_{4}$, for all k≠ℓ. It was assumed that stationary transition rate matrix of η is
$$\begin{array}{c} \Pi =\left[\begin{array}{cc} -0.5 & 0.5\\ 0 & 0 \end{array}\right]. \end{array}$$
The elements of the second row of the stationary transition matrix were taken zero in order to accomplish the condition $X_{2}\left(2\right)=0$. Indeed, for P(2)=0 and for $\pi_{21}=\pi_{22}=0$, the unique stabilising solution of the Riccati system (Equation 21) has the form $\left(X_{2}\left(1\right),0\right)$ since $\left(X_{1}\left(1\right),X_{1}\left(2\right)\right)$ is the stabilising solution of (Equation 20). The elements of the first row of Π have been chosen such that the transition from the nominal first state to the second one corresponding to the network failure takes place in about 5 s as illustrated in Figure 1.

Solving the Riccati systems (Equation 20) and (Equation 21) for γ=100 one obtains
$$\begin{array}{c} \begin{array}{c} X_{1}\left(1\right)=\left[\begin{array}{cccc} 1.7686 & 1.2373 & 0 & 0\\ 1.2373 & 1.7379 & 0 & 0\\ 0 & 0 & 1.7686 & 1.2373\\ 0 & 0 & 1.2373 & 1.7379 \end{array}\right],\\ X_{2}\left(1\right)=\left[\begin{array}{cccc} 204.1418 & 2.0204 & 0 & 0\\ 2.0204 & 1.0006 & 0 & 0\\ 0 & 0 & 204.1418 & 2.0204\\ 0 & 0 & 2.0204 & 1.0006 \end{array}\right],\\ X_{1}\left(2\right)=\left[\begin{array}{cccc} 1.4147 & 1.0002 & 0 & 0\\ 1.0002 & 1.4145 & 0 & 0\\ 0 & 0 & 1.4147 & 1.0002\\ 0 & 0 & 1.0002 & 1.4145 \end{array}\right],\;X_{2}\left(2\right)=0_{4} \end{array} \end{array}$$
resulting the following gains
$$\begin{array}{c} \begin{array}{c} F_{1}\left(1\right)=\left[\begin{array}{cccc} -20.1631 & -10.0793 & 0 & 0\\ 0 & 0 & -20.1631 & -10.0793 \end{array}\right],\\ F_{2}\left(1\right)=\left[\begin{array}{cccc} 0.2024 & 0.1001 & 0 & 0\\ 0 & 0 & 0.2024 & 0.1001 \end{array}\right],\\ F_{1}\left(2\right)=\left[\begin{array}{cccc} -0.1012 & -0.1431 & 0 & 0\\ 0 & 0 & -0.1012 & -0.1431 \end{array}\right],\;F_{2}\left(2\right)=0_{2\times 4}. \end{array} \end{array}$$
Using the aforementioned gains one determined the agents planar trajectories for both cases when the network communication properly works and the case when it fails, respectively. Figure 2 presents two snapshots at *t* = 0.5 s and at *t* = 2 s obtained for random initial positions of the agents in both cases. One can see from the two snapshots in the upper half of the figure, that the matrix gain *P* giving the coupling between agents is important in determining a short settling time. By contrast, numerical simulations show that in the case when the communication network is not available the settling time is about 6.5 s.

## 5. Concluding Remarks

An optimal $H_{\infty }$ type control method for large-scale multi-agent systems with identical dynamics and independently actuated is presented. It is shown that regardless the number of agents, the optimal solution may be obtained solving two systems of algebraic Riccati equations whose dimension correspond to a single agent. Convergent iterative numerical procedures are presented and used for a case study revealing the benefits of the coupling between agents. The proposed design methodology may be also used in applications in which only some of the links between agents fail. The dimension of the Riccati equations remains the same but their number increases due to the larger number of states of the Markov chain. Future research will be devoted to applications in which a sensitivity analysis with respect to the transition probabilities will be included.

## Figures and Tables

**Figure 1 entropy-24-01734-f001:**
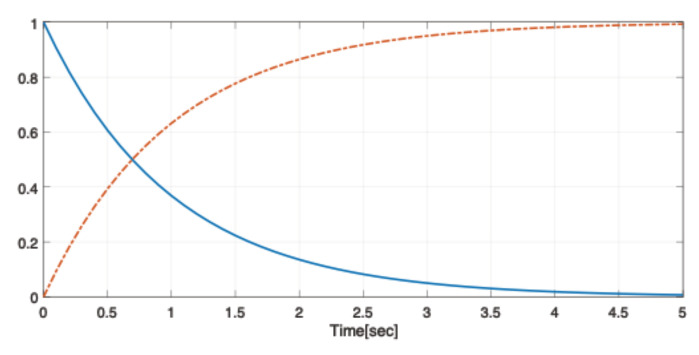
Transition probabilities: $P_{11}\left(t\right)$-solid line, $P_{12}\left(t\right)$-dash-dotted line.

**Figure 2 entropy-24-01734-f002:**
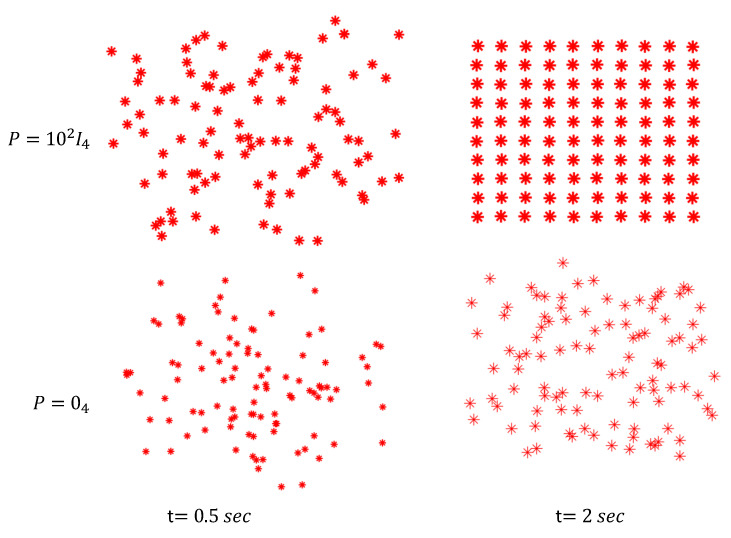
Snapshots for two different coupling weighting matrices of the agents.

## Data Availability

Not applicable.

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
