# Peer review of "H∞ State-Feedback Control of Multi-Agent Systems with Data Packet Dropout in the Communication Channels: A Markovian Approach"

_entropy, 2022, doi:10.3390/e24121734_

Round 1
Reviewer 1 Report
This manuscript is about a Markovian modeled multi-agent system (worst-case) optimization framework in a communication system with potential packet loss. Major observations:
+- Abstract seems to be conveying the idea to some extent. However, the "why" and significance of results components are poor/missing.
-- Introduction refers to many references in a very shallow and unsatisfactory manner. The proposed control related references are definitely inadequate. Citations should be referred individually and concisely about their relevance, importance, linkages, etc., instead of simply lumping them.
-- The assumption of all agents having identical dynamics is an extreme oversimplification of a real problem (at least, individual dynamics and/or stochastic variables are likely different and Remark 2 of the narrative is important) and it is a possible culprit for the claimed result of the Riccati equation order equaling to a single agent dynamical order. At least, bounded model and/or stochastic variable variations need to be explored/included. Also, possible non-identical dynamical system state variable acquisition feasibility, efficiency, etc., also needs to be elaborated.
-+ The mathematical derivations following the literature results can be followed but revision to ensure smooth flow seems to be necessary, e.g., derivations after Line-119.
- Explanations about the matrix assumptions in Line-76-77 are necessary to convince a realistic framework (not the mathematical steps as mentioned in the text).
- The cost function in (11) seems to be missing the integration symbol (based on what is given in (12)).
- Explanation of the stationary transition rate matrix is needed for its values, practical and computational significance, etc.
- Numerical simulation setup, implementations, critical evaluations, etc. seem to be very much limited to show the merit of the proposed framework, if any.
- The numerical convergent algorithm assumes a stabilizable multi-agent system. However, even when considering the identical dynamics for all agents, it is not clear how to ensure this assumption, if possible, with real-life stochastic variations.
- Unexpected presentation issues: "which dimension" on Line-5, unsorted references in the narrative, "these link" on Line-45, "may found" on Line-94, "control low u(t)" on Line-108, "the last above equation" on Line-120, "(24) remain" on Line-214, "above expression of ˜F (2)" on Line-232, "the above gains" on Line-260, "figure that" on Line-263, "which dimension" on Line-271, "For and arbitrary" on Line-297
Author Response
Response for Reviewer 1
Thank you very much for your comments and suggestions.
- We revised the Abstract trying to emphasize the relevance of the results;
- Introduction has been carefully revised and a new version has been written almost entirely, including relevant previous developments and contributions of the paper;
- Indeed, the assumption that all agents have the same dynamics may been restrictive in some applications; however, this assumption is frequently used in many developments considering optimal control of multi-agent systems; some relevant references are also included;
- The derivations after Line 119 have been split for a more fluent reading;
- The integral missing symbol has been inserted in Eq. (11);
- Explanations about the choice of the stationary transition rate matrix have been introduced and some comments have been added in the section presenting the numerical example;
- The proposed iterative algorithm is convergent if the Riccati systems (20) and (21) have stabilising solutions, which means that indeed, the multi-agent system must be stabilisable. The paper did not aim to derive conditions for the existence of such stabilising solutions. Probably this can be an interesting direction of future investigation.
- The language has been carefully checked and the suggested corrections have been made.
Reviewer 2 Report
The paper deals with stabilization, via H_\infty methods, of homogeneous MASs in presence of data packet dropout in the communication channel. Modeling the dynamics as a Markov chain, the design proceeds by solving an optimal stochastic control problem that induces a set of coupled Riccati equations.
The results are interesting and worth publication. However, I have a few comments that might help in highlighting the contribution and the context the paper is settled within.
The title and abstract should mirror the control objective of the paper. This is not the case here as it only highlights the method (H_\infty control) and the context (multi-agent systems with packet dropout). However, what is the control problem in particular ? “Control” is too vague and one should highlight what is the control problem one is facing (e.g., stabilization, tracking, etc).
The literature review may be enriched with further (more recent) developments on multi-agent systems as, for instance, the following ones.
Finally, there are several typos spread all along the manuscript; e.g., line 10 -> “The multi-agent systems … ” should be “Multi-agent systems … ”; line 35 -> “The its order” should be “its order”; line 97; “has a” should be “have a” (the same typo is repeated later on several times); etc
Author Response
Response for Reviewer 2
Thank you very much for your comments and suggestions.
- The title has been slightly changed indicating the optimal control law structure;
- The literature review was almost entirely rewritten presenting in a more detailed manner some previous results in domain;
- The language has been carefully checked and the suggested corrections have been made.